

# Phosphorus dynamics during early soil development in extreme environment

Zuzana Frkova[1,2], Chiara Pistocchi[3], Yuliya Vystavna[2,4], Katerina Capkova[2,5], Jiri Dolezal[5,6], Federica Tamburini[7]

[1] University of Luxembourg, Faculty of Science, Technology and Communication, 6, rue Richard Coudenhove-Kalergi, L-1359, Luxembourg

[2] Biology Centre of the Czech Academy of Sciences, Institute of Hydrobiology, Na Sádkách 7, 370 05 České Budějovice, Czech Republic

[3] Eco&Sols, Montpellier SupAgro, CIRAD, INRAE, IRD, Univ. Montepellier, 34060 Montpellier, France

[4] Biology Centre of the Czech Academy of Sciences, Institute of Soil Biology, Na Sádkách 7, 370 05 České Budějovice, Czech Republic

[5] Czech Academy of Sciences, Institute of Botany v.v.i., Dukelská 135, 379 82 Třeboň, Czech Republic

[6] Faculty of Science, University of South Bohemia, Na Zlaté stoce 1, 370 05, České Budějovice, Czech Republic

[7] Institute of Agricultural Sciences, ETH Zurich, Research Station Eschikon 33, 8315 Lindau, Switzerland

*Correspondence to*: Federica Tamburini (Federica.tamburini@usys.ethz.ch)





**Abstract.** At the early stages of pedogenesis, the dynamics of phosphorus (P) in soils are controlled by microbial communities, the physicochemical properties of the soil and the environmental conditions. While various microorganisms involved in carrying out biogeochemical processes have been identified, little is known about the actual contribution of microbial processes, such as organic P hydrolysis and microbial P turnover, to P cycling. We thus focused on processes driven by microbes and how they affect the size and cycling of organic and inorganic soil P pools along a soil chronosequence in the Chamser Kangri glacier forefield (Western Himalayas). The rapid retreat of the glacier allowed us to study the early stages of soil formation under cold arid climate. Biological P transformations were studied with the help of the isotopic composition of oxygen (O) in phosphate ($\delta^{18}O_P$) coupled to sequential P fractionation performed on soil samples from four sites of different age spanning 0 to 100-150 years. The mineral P, i.e. 1M HCl-extractable P, represented still 95% of the total P stock after approximately 100 years of soil development. Its isotopic composition was similar to the parent material also at the most developed site. Primary phosphate minerals, therefore, mostly composed this pool. The $\delta^{18}O_P$ of the available P and the P bound to Fe and Al oxides instead differed from that of the parent material, suggesting that these pools underwent biological turnover. The isotopic composition of O in of the available P was mostly controlled by the microbial P, suggesting fast exchanges occurred between these two pools possibly fostered by repeated freezing-thawing and drying-rewetting cycles. The release of P from organic P become increasingly important with soil age, constituting one third of the P flux to available P at the oldest site. Accordingly, the lighter isotopic composition of the P bound to Fe and Al oxides at the oldest site indicated that this pool contained phosphate released by organic P mineralization. Compared to previous studies on early pedogenesis under alpine or cold climate, our findings suggest a much slower decrease of the P-bearing primary minerals during the first 100 years of soil development under extreme condition. However, they provide evidence that, by driving short-term P dynamics, microbes play an important role in controlling the redistribution of primary P into inorganic and organic soil P pools.

## 1 Introduction

Microorganisms, together with climate, relief, parent material and time, drive the transformation of chemical elements, and thus nutrients like phosphorus (P), in the early stages of soil formation (Paul and Clark, 1989; Dietrich and Perron, 2006; Egli et al., 2011). The amount of inorganic nutrients released depends on C availability, the mineral nutrient demand of the microorganisms, and the nutrient content of the substrate (Mooshammer et al., 2014)(Moorhead et al., 2012).

During the early stages of pedogenesis, inorganic P (Pi) from the parent material accounts for most of soil total P and progressively declines, while available P (i.e. readily utilized by plants or microorganisms) initially increases due to weathering of P-bearing minerals. In later stages of soil development, Pi decreases as P becomes sequestered in organic forms, occluded in secondary minerals or is lost via leaching/erosion (Walker and Syers, 1976). Organic P (Po) might rapidly become an important source of available P through the mediation of soil microorganisms (Egli et al., 2012). Wang *et al.* (2016) showed that Po mineralization was coupled to carbon (C) mineralization in a glacier forefield and



hypothesized that Po mineralization was driven by the microbial need of C. The microbial P pool could also contribute to available P via the turnover of microbial cells determined by biological (e.g. grazing) or abiotic (e.g. freezing/thawing) factors (Oberson and Joner, 2005). However, little is known about the actual contribution of microbial processes to P availability especially during the early pedogenesis (Schulz et al., 2013).

The study of isotopic composition of oxygen (O) in phosphate (expressed here as $\delta^{18}O_P$ in the delta per mil notation) is a relatively new approach to trace P biogeochemical transformation processes overcoming limitations of approaches that depend on radioisotopes (Frossard et al., 2011). The P-O bond in phosphate is resistant to inorganic hydrolysis under natural temperature and pressure (O'Neil et al., 2003), therefore negligible O atoms exchange occurs between phosphate and water without biological mediation (Tudge, 1960; Blake et al., 2001). Two main enzyme-mediated processes can alter the $\delta^{18}O_P$. First, intracellular metabolism of Pi (reversible conversion of pyrophosphate to two phosphate ions mediated by inorganic pyrophosphatase) causes a complete and fast exchange of O atoms between phosphate and water molecules (Cohn, 1958; Blake et al., 2005). This results in a temperature-dependent equilibrium between O in the phosphate molecule and O of the intracellular water (Fricke et al., 1998). The $\delta^{18}O_P$ at equilibrium can be predicted, based on the measured isotopic ratio of oxygen in water ($\delta^{18}O_W$) and temperature (Longinelli and Nuti 1973; Kolodny *et al.* 1983; Chang and Blake 2015). Therefore, the difference between the calculated equilibrium value and the measured $\delta^{18}O_P$ in an environmental sample can provide insights into the extent to which Pi has been cycled by the soil microorganisms through intracellular metabolic reactions (Davies et al., 2014). The second process is the hydrolysis of Po compounds by phosphohydrolases-mediated reactions (Liang and Blake, 2006). During Po hydrolysis, P-O bonds are cleaved and one to two O atoms in the phosphate are replaced with O from the water with a specific fractionation factor ($\varepsilon$). The $\varepsilon$ of main phosphohydrolase enzymes is often negative, e.g. alkaline phosphatases have an $\varepsilon$ of -30‰, while acid phosphatases of -10‰, resulting in depleted $\delta^{18}O_P$ in the released phosphate (Liang and Blake, 2009; Von Sperber et al., 2015).

Glacier forefields are ideal sites to study the initial steps of soil formation as neighboring sites represent a soil chronosequence of different soil developmental stages. The general assumption is the space for time substitution, implying that each site along a glacier forefield chronosequence was subject to the same initial conditions and followed the same sequence of changes. We studied the early stages (0 to approximately 100-150 years) of soil development in a glacier forefield located in the Western Himalayas. Under these conditions, microorganisms may be subjected to drought, intense solar radiation (Blumthaler et al., 1997), and high temperature fluctuations (Janatková et al., 2013; Rehakova et al., 2011). Direct forefield observations on the role of microorganisms in P cycling using O isotopes in phosphate are rare. A study in the forefield of the Damma glacier in the Swiss Alps revealed that at the early stages of



soil development (<150 years), the microbial P was the main contributor to available P, which carried the isotopic

equilibrium signature (Tamburini et al., 2012). More recently, it has been shown that under P-limiting conditions,

instead, available P might show non-equilibrium lower $\delta^{18}O_P$ as a result of tight Po recycling through phosphohydrolase-

mediated reactions (Pistocchi et al., 2020).

In this study, we applied a sequential fractionation method to identify loosely to strongly bound Pi and Po pools and

analyzed the $\delta^{18}O_P$ values in these pools (Tamburini et al., 2018) to quantify biological P contribution to available P

during early soil formation. Given that sequentially-extracted P pools differ in their availability and turnover time

(Helfenstein et al., 2020), we hypothesized 1) that the influence of biological cycling will increase with increasing soil

age; 2) that the main contribution to available P, at the early development stages, will derive from microbial P rather

than Po mineralization; and 3) that the contribution to available P from Po mineralization will increase with soil age as

observed in an older chronosequence (Roberts et al., 2015). We will therefore: i) quantify sequentially extracted P pools

and potential enzymatic activities linked to C, nitrogen (N) and P mineralization in soils along the forefield

chronosequence, ii) determine the $\delta^{18}O_P$ values of these pools, including Po and compare them to the calculated isotopic

equilibrium and expected isotopic values from Po hydrolysis, in order to assess their relative contribution to available

P.

## 2 Materials and methods

### 2.1 Study site description

The glacier forefield of Chamser Kangri was chosen as it has been studied within a long-term interdisciplinary research

development project (Dolezal et al., 2016). Since 2008, the local environmental conditions have been monitored,

changes in vegetation and relationships between vascular plants and soil microbial communities studied (Dvorský et al.,

2015, 2011; Janatková et al., 2013; Řeháková et al., 2017; Rehakova et al., 2011; Čapková et al., 2016; Aschenbach et

al., 2013). The Chamser Kangri glacier is located in Ladakh, Northwest India, in the southwestern extension of the

Tibetan Plateau on the northern slope of Chamser Kangri peak (6645 m a.s.l.) belonging to the Lungser Range above

Tso Moriri lake. The glaciated area of the Lungser Range decreased from 61.2 km$^2$ in 1969 to 55.4 and 50.4 km$^2$ in

years 2003 and 2014, respectively. The estimated average retreat of the glacier based on the measurements in 39 valleys

indicated 2.6 m per year between 1969 and 2003, and almost the double between 2003 and 2014 (Schmidt and Nüsser,

2017). The climate in mountain areas of the Northwest Himalayan region, at an altitude of >5000 m a.s.l., is arid and

characterized by annual precipitation in the range of 50-150 mm y$^{-1}$, rarely affected by monsoonal precipitation

(Customized Rainfall Information System (CRIS) - Hydromet Division 2012-2016; Harris, 2006).



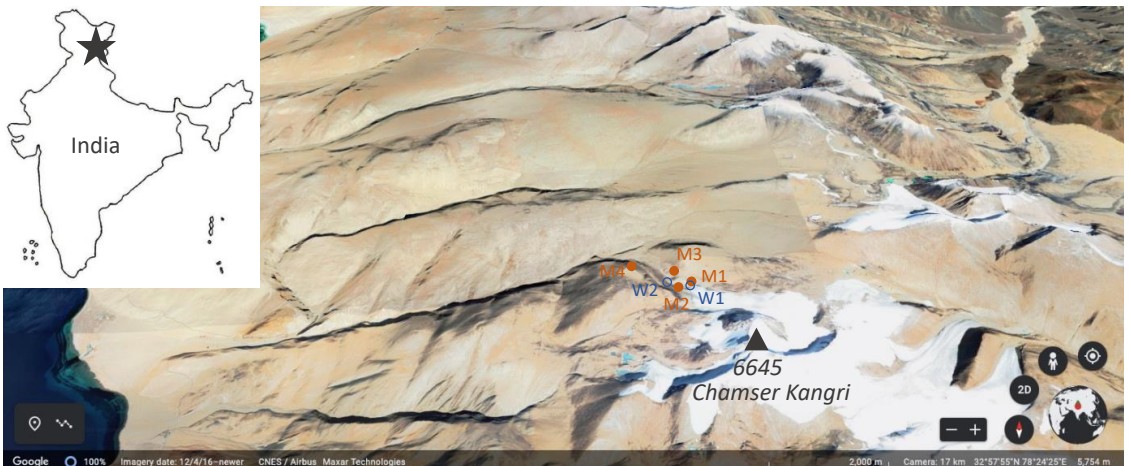

Figure 1: Chamser Kangri glacier forefield chronosequence in India (adapted outline map from coloringhome.com)
with sampling sites (M for soils and W for stream water and glacier snout) indicated on a satellite map (35°57'55" N
78°24'25" E. © Google Earth, CNES / Airbus Maxar Technologies, April 12 2016). Orange and open blue rings
designated sites of soil and water, respectively.

The soils had a coarse-grained structure, with 30-50% of gravel and a pH between 7.8 and 8.5 (Rehakova et al., 2011).
The parent rock consists mainly of metasedimentary siliceous rocks, the Tso Morari granite gneiss (Epard and Steck,
2008). The cold steppe vegetation cover is sparse, characterized by alpine grasslands mainly constituted by
hemicryptophytes (Dvorský et al., 2015, 2011). The environmental conditions give rise to an extensive development of
soil biological crusts dominated by cyanobacteria, which cover up to 40% of the soil surface in this area, and facilitate
the initial establishment of vascular plants (Čapková et al., 2016; Řeháková et al., 2017). Cyanobacterial communities
(*Nostocales*, *Chroococcales* and *Oscillatoriales*) were observed to be more abundant and species-rich in bare than
vegetated soils (Řeháková et al., 2017). Together with Cyanobacteria, Gram-positive spore-forming bacteria
(*Actinobacteria* and *Firmicutes*) were the main recorded bacterial clades (Řeháková et al., 2015).

### 2.2. Sampling description

Soil samples were collected along the frontal (M1-3) and lateral (M4) moraines of the Chamser Kangri glacier
chronosequence (5711, 5710, 5700 and 5598 m a.s.l., M1-M4 respectively). The age of the chosen soils was estimated
to be <1, 25, 50 and > 100 years, for M1 to M4, respectively, according to the slope and relative ice cover loss of 0.4%
km$^{-2}$ year$^{-1}$ (Dvorský et al., 2015). Sites M1 and M2 had no vegetation cover, M2 and M3 had weakly and well-developed
soil biological crusts, respectively, M3 was sparsely covered by *Thylacospermum caespitosum* (*Caryophyllaceae*



family) plants, and M4 had well-developed soil biological crusts and vegetation cover of *Poa attenuata* and *T. caespitosum* (Figure S2). The average soil temperature (5 cm below surface) during the week (23/2017) of sampling was 0.18°C, which corresponded to previous years values (based on long-term data collected from installed microclimatic stations TOMST®, Figure S1A). The average soil moisture for the corresponding week was around 0.25 $cm^3 \, cm^{-3}$ and was 25-42% higher in respect to the corresponding period in the previous years (Figure S1B). The sampling

period represented the first moist period with above-zero temperatures after a long dry spell (Figure S3).

Representative samples were obtained by pooling ~100 g of upper soil surface 0-5cm, collected at 10 different spots, i.e. approximately 1 kg of soil, at each locality (M1-4, Figure 1) along the glacier forefield chronosequence.

To evaluate the hydrological and evaporation influence on water oxygen ($\delta^{18}O_W$) and hydrogen ($\delta^2H_W$) isotopic composition, samples of the glacier snout and the stream water (covered by ice) coming from the glacier (W1-2, Figure

1), were taken. Samples were stored in zip-lock plastic bags/HDPE bottles and a thermo-bag to minimize biological activity during transport to the laboratory. Soils were sieved after removing the gravel material (2-mm mesh) and homogenized. Subsamples for cryo-distillation, dry weight measurement and enzymatic activities were taken and stored at 4°C until processed.

**2.3 General soil characteristics**

The general soil characteristics were measured using standard laboratory procedures on the sieved samples. Soil pH was measured in 1:5 soil: water suspensions (ISO, 2005), total organic carbon and total nitrogen were measured on a TOC/TN analyser (Formacs), total P and micronutrients were analyzed by ICP-OES after sequential digestion by $HNO_3$ and $HClO_4$ (Kopacek et al., 2001). Soil texture was estimated based on particle size analysis (Afnor, 1994).

Bulk density of soil, used to calculate P stocks, was estimated using a pedotransfer function (Leonaviciute, 2000), corresponding to eluvial deposits considering both the soil texture and organic carbon content.

**2.4 Extraction and purification of phosphate in different pools**

Phosphorus was sequentially extracted from the soil following a modified Hedley fractionation, which was upscaled to

obtain a sufficient amount of phosphate for isotopic analyses according to (Tamburini et al., 2018). Five different P pools were defined: resin- and hexanol-extractable P (bioavailable and microbial, respectively), NaOH-EDTA extractable P (bound to iron- (Fe) and aluminum- (Al) oxides and to organic P), and HCl extractable P (mineral P, mostly P bound to Ca in apatite). Additionally, parent material samples were crushed and extracted directly with 1M HCl. To measure the $\delta^{18}O_P$ values, all soil extracts were purified following the protocols of Tamburini et al. (2010,



2018). The inorganic P concentration in each pool was determined by the malachite green method (Ohno and Zibilske,

1991). For some samples, the soil available was not enough to extract and purify enough P to measure $\delta^{18}O_P$ reliably for

all pools. This was the case of microbial P in the samples from sites M1 and M2 and the P bound to oxides in the sample

from site M2. Only the organic pool of sample M4 contained enough material for $\delta^{18}O_P$ analyses.

**2.5 Stable isotope analysis of oxygen in water and phosphates**

Soil water was extracted by cryogenic vacuum extraction from one sample per sampling site (Orlowski et al., 2013).

Analyses of stable isotopes in samples from glacier snout and glacier stream water ($\delta^{18}O_W$ and $\delta^2H_W$) were performed

using the L2120i laser instrument (Picarro Inc.) at the Institute of Hydrology, Slovakia. Hydrogen and oxygen isotope

analyses were calibrated against the V-SMOW (Vienna Standard Mean Ocean Water) and were reported in the standard

‰ notation. Typical precisions are better than ±0.1‰ for both $\delta^{18}O_W$ and $\delta^2H_W$, respectively. The oxygen isotopes in

soil water were measured by equilibration with $CO_2$ (Seth et al., 2006) using a gas bench (Thermo Scientific Gas Bench

II) connected to an isotope ratio mass spectrometer (Thermo Scientific Delta V plus) at the Stable Isotope Laboratory

of the Geological Institute of the ETH Zurich. The system was calibrated against V-SMOW, SLAP (Standard Light

Antarctic Precipitation) and GISP (Greenland Ice Sheet Precipitation).

To distinguish between evaporated and non-evaporated stream water sources, we used deuterium excess (d-excess) that

is associated with kinetic isotopic fractionation and calculated as d-excess = $\delta^2H - 8 * \delta^{18}O$ (Dansgaard, 1964). Samples

with d-excess value <10 suggest a deviation from the equilibrium fractionation conditions, indicating that the water may

have been subject to evaporation (Dansgaard, 1964).

The stable oxygen isotope signature in phosphate was determined on a Vario Pyro Cube (Elementar, GmbH, Hanau,

Germany) coupled in a continuous flow to an Isoprime 100 isotopic ratio mass spectrometer (Isoprime, Manchester,

UK). Calibration and corrections for instrumental drifts were done by repeated measurements of a $Ag_3PO_4$ internal

standard (with a value of +14.20‰) and benzoic acids IAEA 601 and 602. The $\delta^{18}O$ values are expressed in the standard

delta notation with respect to V-SMOW. Reproducibility of the measurements based on repeated measurements of the

internal standard was within 0.4‰.


**2.6 Calculations of isotopic equilibrium values and box model**

The equilibrium between oxygen in phosphate and water, given by intracellular P turnover by pyrophosphatase (PPase)

was computed using the revised Chang and Blake equation (Chang and Blake, 2015):



$$\delta^{18}O_{P(eq)} = -0.18\ T + 26.3 + \delta^{18}O_W$$

(1)

where $T$ is the temperature in °C and $\delta^{18}O_W$ is the isotopic composition of water in ‰. To integrate the variability at different time scales, Eq. 1 was solved using values from two reference periods: i) the 10 days preceding the sampling (from June 6, 2017, hereafter equilibrium 1), using on site soil temperatures and the $\delta^{18}O_W$ of soil water as the upper limit and $\delta^{18}O_W$ of June monthly rainfall adjusted for the altitudinal gradient as by Lone (2019) as lower limit; ii) a

multi-annual average (hereafter equilibrium 2), calculated using a temperature range spanning the mean soil temperature of warm months (years 2013-2017) and 0°C, considering biological activity as negligible below 0°C. We took a weighted average of seasonal precipitations, again adjusted for altitude (Lone 2019) with or without the evaporative enrichment observed in soil water. In using rainfall $\delta^{18}O_W$ as a proxy for soil water isotopic composition, we assumed that soil water reflects a mass balance of seasonal precipitation short of an evaporative enrichment (Roberts et al., 2015;

Sprenger et al., 2017).

The expected $\delta^{18}O_P$ values of phosphate mineralized from phosphomonoesters (PME, $\delta^{18}O_{PME}$) were calculated as follows (Liang and Blake, 2006):

$$\delta^{18}O_{PME} = 0.25\ (\delta^{18}O_W + \varepsilon) + 0.75\ \delta^{18}O_{P\text{-org}} \qquad (2)$$

using the fractionation factor ($\varepsilon$) of -30‰ for hydrolysis of Po by alkaline phosphatases, as the pH range of studied soils

is alkaline (Table 1), and a $\delta^{18}O_{P\text{-org}}$ value of +12.84‰, as the Po pool in site M4 assuming similar values for the other soils.

To determine the contribution to available P from the mineral, organic and microbial P pools, we estimated P fluxes and calculated the expected isotopic composition of the available P as follows (box model approach, Tamburini et al., 2012):

$$\delta^{18}O_{Pexpected} = (f\text{-}_{Pmic}\ \delta^{18}O_{Pmic} + f\text{-}_{Po}\ \delta^{18}O_{PME} + f\text{-}_{Pmineral}\ \delta^{18}O_{Pmineral})\ /\ (f\text{-}_{Pmic} + f\text{-}_{Po} + f\text{-}_{Pmineral}) \qquad (3)$$

where $f\text{-}_{Pmic}$ is the P flux from microbial P turnover (mg P m$^{-2}$ day$^{-1}$, see below) and $\delta^{18}O_{Pmic}$ the isotopic composition of the microbial P; $f\text{-}_{Po}$ is the P flux from the mineralization of Po, $\delta^{18}O_{PME}$ the expected $\delta^{18}O_P$ values of released phosphate; and $f\text{-}_{Pmineral}$ is the flux from the mineral P and $\delta^{18}O_{Pmineral}$ the corresponding isotopic composition.

When the expected isotopic signatures matched the measured ones, we assumed P fluxes were estimated correctly. The flux from the hydrolysis of Po was estimated both from microbial respiration rates measured at 10°C in the same area

(comparable period of sampling, distance to the glacier, age; pers.com. Capkova et al.) and from potential phosphatase activities (Phillips et al., 2005) (Table 2). The first approach assumes that the mineralized P is proportional to the organic C released as $CO_2$, according to their stoichiometric ratio in non-living organic matter (Achat et al., 2010; Bünemann, 2015):



$$f_{Po} = \frac{C\text{-}CO_2 / \frac{(1 - microbial\ C\ efficiency)}{C:Po}}{} AM_P \qquad (4)$$

Where $C\text{-}CO_2$ is the carbon released through soil respiration (µmoles C m$^{-2}$ day$^{-1}$) adjusted for the average soil

temperature during the sampling period according to Lloyd and Taylor (1994), microbial efficiency was set at 0.4

(Murphy et al., 2003), C:Po is the molar ratio between the Po (0.25M NaOH-0.05M EDTA extraction) and the soil

organic carbon and $AM_P$ the atomic mass of P.

The second approach yielded very high Po mineralization rates and consequently the $\delta^{18}O_{Pexpected}$ of available P were

strongly depleted compared to the measured values. This confirms that measured phosphomonesterase activities

reflected potential rather than actual rates (Bünemann, 2015). This second approach was, therefore, discarded. To

estimate P fluxes from microbial P (f-$_{Pmic}$), we considered two different turnover times for microbial P: 15 days (as

reported for microbial N and microbial P in P-depleted forest soil, Fisk et al. 1998; Schmidt et al. 2007)(Pistocchi et al.,

2018) and 70 days (as reported for microbial P in arable soils, Oehl et al. 2001). Finally, the flux from the mineral P (f-

$_{Pmineral}$) was estimated by dividing the difference in stock concentrations by the difference in age of sites M4 and M1, as

in Tamburini et al. (2012).

To account for uncertainties introduced with calculations, the mean and standard deviation of the expected isotopic

signatures were obtained with a Monte Carlo error propagation simulation (Anderson, 1976). Calculations were repeated

10 million times by varying the variables ($\delta^{18}O$ and P fluxes) according to their mean and standard deviation from

analytical replicates. When the $\delta^{18}O_P$ of microbial P were missing (site M1 and M2), we used the entire range of values

measured at sites M3 and M4.

**2.7 Soil enzyme activities**

To understand the regulation of enzymes activity in our soil chronosequence, we assessed five enzymes involved in

mineralization of organic C, N and P (β-glucosidase, cellobiosidase, chitinase, leucine aminopeptidase and acid/alkaline

phosphatase) (Marx et al. 2001; Bárta et al. 2014). Briefly, 0.5 g of soil was homogenized in 10 mL of MQ water using

an IKA Ultra-Turrax T 10 homogenizer (IKA-Werke GmbH & Co. KG, Germany). Soil suspensions of 200 µL were

transferred to a 96-well microplate in four analytical replicates. Then 50 µL of corresponding labelled substrate

according to the soil enzyme (4-methylumbelliferyl-β-D-glucopyranosidase, 4-methylumbellyferyl-N-

cellobiopyranoside, 4-methylumbellyferyl-N-acetglucosaminide, L-leucine-7-amido-4-methylcoumarin or 4-

methylumbellyferyl-phosphate) were added. Microplates were incubated at 30°C. Fluorescence time-related

measurements were performed after 30, 90 and 150 min on an INFINITE F200 (TECAN, Crailsheim, Germany)



microplate reader using the excitation and emission wavelength of 365 and 450 nm, respectively. Enzyme efficiencies were calculated as enzyme activities in nmol $h^{-1}$ $g^{-1}$ soil dry weight, using an eight-point calibration curve.

## 3 Results

### 3.1 General soil characteristics

Particle size analysis revealed all forefield soils being sandy, with 78.0-85.5% of sand (Table 1). M1 had the highest portion of clay, silt and fine sand, 4.1, 17.8 and 58.6%, respectively. The soil at the M3 site was the poorest in clay with only 0.5%. The estimated bulk density ranged between 1.4 and 1.7 Mg $m^{-3}$ from the oldest soil to the youngest,

respectively. The total organic carbon concentration increased with soil age, from 0.6 g $kg^{-1}$ in the youngest soil M1 to 21.7 g $kg^{-1}$ at M4. The soil N content was low in the young soils at M1-M3, ranging from 0.03 to 0.6 and was higher at M4 with 2.4 mg $kg^{-1}$. The total P did not show a particular trend along the chronosequence, ranging from 0.72 to 0.93 g $kg^{-1}$, with the lowest content at M2 and highest at M3.

The molar C:N ratio was relatively constant along the chronosequence, averaging 9, except at the youngest site M1,

where total nitrogen concentration was very low and the C:N ratio the highest (Table 1). The C:Po and N:Po molar ratios increased along the chronosequence from 527 to 932 and from 24 to 86, respectively. This increase was greater between 0 and 25 years (M1 and M2) and 50 and 100 years (M3 and M4).

The soil pH gradually decreased from 8.7 at the youngest site M1 to 7.7 at the oldest site M4.



Table 1: General characteristics of the soil top 5 cm along the glacier forefield chronosequence (n=3): soil age, texture,

bulk density, pH, nutrient content, C:N:P stoichiometric ratios, soil P pools, and enzyme activities.

| Parameters | Sites | | | | | | | |
|---|---|---|---|---|---|---|---|---|
| | M1 | | M2 | | M3 | | M4 | |
| | average | stdev | average | stdev | average | stdev | average | stdev |
| Soil age | <1 | n.a. | 25 | n.a. | 50 | n.a. | >100 | n.a. |
| Clay [%] | 4.11 | n.a. | 1.57 | n.a. | 0.49 | n.a. | 2.51 | n.a. |
| Fine silt [%] | 0.88 | n.a. | 1.23 | n.a. | 1.60 | n.a. | 2.16 | n.a. |
| Coarse silt [%] | 16.93 | n.a. | 12.51 | n.a. | 15.44 | n.a. | 9.86 | n.a. |
| Fine sand [%] | 58.59 | n.a. | 44.32 | n.a. | 41.96 | n.a. | 45.64 | n.a. |
| Coarse sand [%] | 19.49 | n.a. | 40.37 | n.a. | 40.52 | n.a. | 39.83 | n.a. |
| Bulk density [mg m$^{-3}$][a] | 1.73 | 0.001 | 1.56 | 0.003 | 1.39 | 0.001 | 1.41 | 0.003 |
| pH | 8.71 | 0.020 | 8.09 | 0.270 | 7.82 | 0.175 | 7.73 | 0.085 |
| TOC [g kg$^{-1}$] | 0.63 | 0.090 | 2.12 | 0.068 | 5.41 | 0.096 | 21.67 | 0.750 |
| TN [g kg$^{-1}$] | 0.03 | 0.003 | 0.29 | 0.016 | 0.61 | 0.008 | 2.34 | 0.042 |
| TP [g kg$^{-1}$] | 0.74 | 0.004 | 0.72 | 0.058 | 0.93 | 0.020 | 0.77 | 0.025 |
| C:N:P molar ratio | 22.3:0.1:1 | n.a. | 8.6:0.9:1 | n.a. | 10.3:1.5:1 | n.a. | 10.8:6.7:1 | n.a. |
| C:Po molar ratio | 527 | n.a. | 695 | n.a. | 648 | n.a. | 932 | n.a. |
| N:Po molar ratio | 24 | n.a. | 81 | n.a. | 63 | n.a. | 86 | n.a. |
| Fe [g kg$^{-1}$] | 11.68 | 0.330 | 12.51 | 0.531 | 11.43 | 0.351 | 14.43 | 0.282 |
| Al [g kg$^{-1}$] | 14.51 | 0.790 | 13.39 | 0.366 | 12.74 | 0.206 | 15.06 | 0.237 |
| Ca [g kg$^{-1}$] | 4.00 | 0.076 | 3.78 | 0.002 | 3.67 | 0.179 | 9.44 | 0.270 |
| K [g kg$^{-1}$] | 8.45 | 0.458 | 7.74 | 0.389 | 6.81 | 0.157 | 7.35 | 0.166 |
| Mg [g kg$^{-1}$] | 4.37 | 0.219 | 4.97 | 0.223 | 4.07 | 0.118 | 5.66 | 0.165 |
| Available P stock [g m$^{-2}$] | 0.03 | 0.002 | 0.08 | 0.013 | 0.16 | 0.017 | 0.16 | 0.032 |
| Microbial P stock [g m$^{-2}$] | 0.04 | 0.002 | 0.11 | 0.019 | 0.25 | 0.027 | 0.50 | 0.103 |
| Oxides-P stock [g m$^{-2}$] | 0.20 | 0.014 | 0.53 | 0.090 | 0.58 | 0.062 | 0.63 | 0.130 |
| Organic P stock [g m$^{-2}$] | 0.27 | 0.02 | 0.62 | 0.08 | 1.50 | 0.45 | 4.22 | 2.78 |
| Mineral P stock [g m$^{-2}$] | 63.81 | 1.443 | 55.61 | 0.943 | 63.35 | 1.677 | 51.58 | 1.06 |
| β-glucosidase [nmol h$^{-1}$g$^{-1}$] | 0.50 | 0.122 | 37.50 | 3.114 | 38.97 | 6.856 | 155.45 | 9.12 |
| Cellobiosidase [nmol h$^{-1}$g$^{-1}$] | 0.15 | 0.070 | 3.76 | 0.982 | 2.51 | 0.981 | 9.18 | 0.500 |
| Leu-aminopeptidase [nmol h$^{-1}$g$^{-1}$] | 9.47 | 0.572 | 66.42 | 5.096 | 101.97 | 3.196 | 179.79 | 7.408 |



| | | | | | | | | |
|---|---|---|---|---|---|---|---|---|
| Chitinase [nmol h$^{-1}$g$^{-1}$] | **0.24** | 0.081 | **3.15** | 0.648 | **8.74** | 3.070 | **9.76** | 0.759 |
| Phosphatases [nmol h$^{-1}$g$^{-1}$] | **4.08** | 2.499 | **73.21** | 11.546 | **78.58** | 17.598 | **208.88** | 7.985 |

$^a$ derived according to (Leonaviciute, 2000)



### 3.2 Enzymes activity

The activity of all enzymes involved in C, N and P mineralization increased with site age ($R^2$ = 0.91, 0.97 and 0.88 respectively; Table 1, Table S2). The sum of the C-decomposing (β-glucosidase and cellobiosidase) and N-decomposing enzymes (leucine aminopeptidase and chitinase), was positively correlated with TOC and TN, respectively ($R^2$ = 0.97 and 0.93). Phosphatase activities were positively correlated with activities of other enzymes, and several soil characteristics (TOC, TN, fine silt content, total P, Ca, Fe, Mg). The activities increased with soil age, and were inversely correlated with the HCl extracted mineral P and positively with all other P pools (Table S2). Overall, the lowest activity was measured for chitinase and cellobiosidase enzymes responsible for the hydrolysis of glycosidic bonds in chitin and cellulose, respectively, ranging from 0.2 to 9.8 nmol $h^{-1}$ $g^{-1}$ of soil. The highest enzyme activity was measured for phosphatases, with 208.9 nmol $h^{-1}$ $g^{-1}$ at M4. Despite the increasing activity from less to more developed soils, very similar activities were observed at M2 and M3 for β-glucosidase and phosphatases, enzymes responsible for the hydrolysis of glucose from cellobiose, and of phosphate from phosphosaccharides, nucleotides and phospholipids, respectively (Table 1).

### 3.2 Stable isotopes in glacier snout, stream and soil water and isotopic equilibrium

The isotopic values at the glacier snout were -15.8‰ for $\delta^{18}O_W$ and -116.3‰ for $\delta^2H_W$, and -12.4‰ for $\delta^{18}O_W$ and -94.3‰ for $\delta^2H_W$ in the glacier stream water. The d-excess at the glacier snout was 10.1% suggesting that there was no evaporation bias. However, d-excess for the stream water was 4.9%, which is indicative of a strong evaporation signal. The $\delta^{18}O_W$ values of the soil water from the four sites were variably enriched compared to the glacier snout. They ranged from -12.6‰ for site M1, located the glacier snout to around -3.0‰ and -5.6‰ for sites M2-M3 and M4, respectively, more distant from the retreating glacier (Table 2).

The $\delta^{18}O_{P(eq)}$ values expected at equilibrium (Eq. 1) ranged from +9.8‰ to +23.4‰ for the short-term equilibrium and from +9.0 to +23.4‰ for the long-term equilibrium (Table 2). The upper range of the isotopic equilibrium varied according to the $\delta^{18}O_W$ values of the soil water and was therefore significantly higher at the intermediate sites M2-M3 than that at sites M1 and M4, due to heavier $\delta^{18}O_w$ (Table 2).

Table 2: Isotopic equilibrium values $\delta^{18}O_{P(eq)}$ calculated according to Eq. 1. Short-term represents the sampling period ten-days equilibrium. Long-term represents a multi-annual equilibrium average.

| | site | $\delta^{18}O_w$ range | | temperature range | | $\delta^{18}O_{P(eq)}$ range | |
|---|---|---|---|---|---|---|---|
| | | ‰ | | °C | | ‰ | |
| Short-term | M1 | -12.63 (0.25) | -16.3 | 0 | 1 | 13.7 | 9.8 |
| | M2 | -3.07 (0.18) | -16.3 | 0 | 1 | 23.2 | 9.8 |

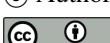



| | | | | | | | |
|---|---|---|---|---|---|---|---|
| | M3 | -2.89 (0.09) | -16.3 | 0 | 1 | 23.4 | 9.8 |
| | M4 | -5.55 (0.11) | -16.3 | 0 | 1 | 20.8 | 9.8 |
| Long-term | M1 | -12.63 (0.25) | -16.5 | 0 | 5 | 13.7 | 9.0 |
| | M2 | -3.07 (0.18) | -16.5 | 0 | 5 | 23.2 | 9.0 |
| | M3 | -2.89 (0.09) | -16.5 | 0 | 5 | 23.4 | 9.0 |
| | M4 | -5.55 (0.11) | -16.5 | 0 | 5 | 20.8 | 9.0 |


### 3.3 Phosphorus soil pools concentrations and their oxygen isotopic value

The sequentially extracted P pools are shown in Table S1. The mineral P pool accounted for 95.1 to 99.5% of total P. The concentrations of the other pools were low and increased with soil age. The available P ranged between 0.3 and 2.3

mg P kg$^{-1}$, the microbial P between 0.4 and 7.1 mg P kg$^{-1}$, the P bound to Fe- and Al-oxides increased from 2.3 to 9.0 mg P kg$^{-1}$, and the organic P from 3.1 to 60.0 mg P kg$^{-1}$.

The $\delta^{18}O_P$ of the available P ranged from +4.66‰ at site M1 to +8.90‰ at site M3. The isotopic value of the microbial P was higher compared to the available P and equal to +12.9‰ at site M3 and +7.63‰ at site M3 (Figure 2). The mineral P pool displayed the closest value to the isotopic composition of the parent material and varied very little between +7.95

and +8.72‰, with the exception of site M3, where it was slightly lower compared to the parent material signature. The P bound to Al- and Fe-oxides showed isotopic values similar to the parent material except for M3 and M4, where this pool carried a $\delta^{18}O_P$ of +15.40‰ and +6.05‰, respectively. The Po pool at site M4 had a $\delta^{18}O_P$ of +12.84‰. Most of the measured $\delta^{18}O_P$ values did not fall within the short-term isotopic equilibrium with ambient water.





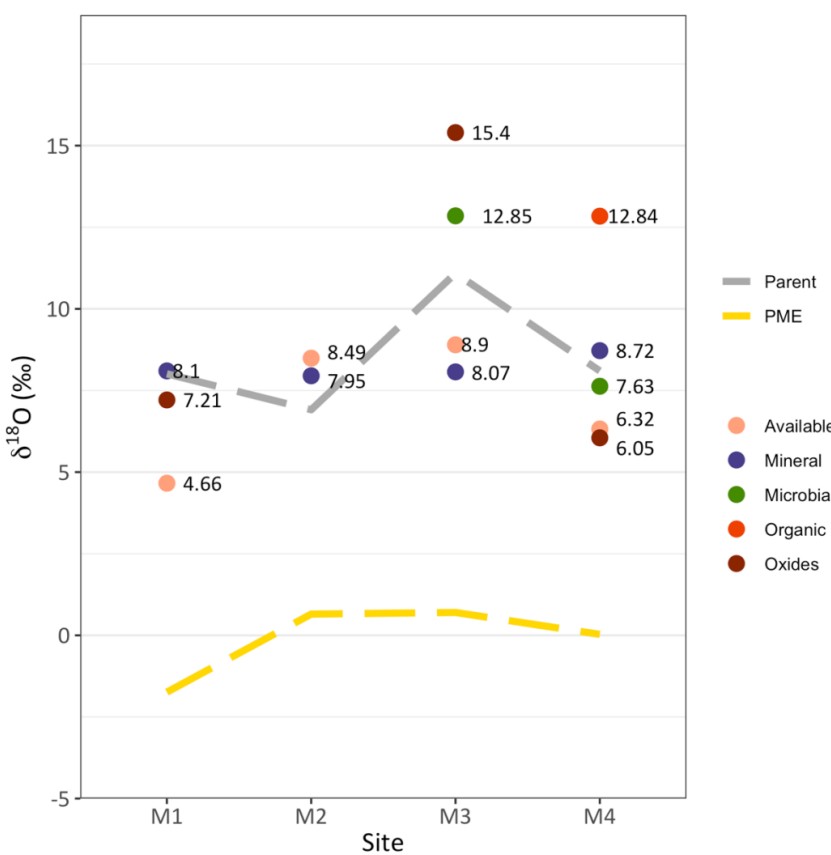

Figure 2: Isotopic values of oxygen in phosphate ($\delta^{18}O_P$) in different soil P pools along the Chamser Kangri glacier forefield chronosequence (<1, 25, 50 and > 100 years, for M1, M2, M3 and M4, respectively). The expected $\delta^{18}O$ value of phosphate derived from Po hydrolysis due to phosphomonoesterases (PME, $\delta^{18}O_{PME}$, Eq. 2) is shown by the yellow discontinuous line, while the grey discontinuous line stays for the isotopic value of the parent material.

## 4 Discussion

### 4.1 Nutrients limitation and microbial P dynamics during early stages of pedogenesis

Nitrogen is absent from most parent materials, but increases across the stages of ecosystem development through biological $N_2$ fixation (Smittenberg et al., 2012). Increased C and N availability along with soil development may eventually result in P becoming the secondary limiting nutrient (Göransson et al., 2011).

Accordingly, total N concentration in the Chamser Kangri forefield increased with soil age, along with the potential activity of C- and N-decomposing enzymes. The C:N ratio, consequently, decreased (Table 1). While (Kaye and Hart 1997) found a shift from C to N limitation at a C:N ratio between 8 and 16, (Göransson et al., 2011) reported a clear N





limitation in most soils with a C:N ratio above 13. We concluded that the most limiting nutrient to primary production

at the youngest site was N (cf. C:N:P ratios in Table 1). This conclusion is supported by the occurrence of N-fixing

cyanobacteria in soil biological crusts at the younger sites (Couradeau et al., 2016). These N-fixing species could provide

with a significant N input in subnival soils (Janatková *et al.* 2013).

The increasing trend of the C:Po and N:Po molar ratios indicates that C and N accumulated faster than P in the soil

organic matter (Table1). On the pedogenesis time scale, assuming inputs to soil organic matter to be mainly derived

from microbial products, C and Po accumulate at a similar rate when P is not limiting and the mineralization of Po is

driven by the need for carbon rather than the need of P (Wang et al., 2016). The increase in C:Po and N:Po ratios may,

therefore, suggest the progressive onset of P limitation or co-limitation. Alternatively, it may point to a shift in the

quality (stoichiometry, recalcitrance) of organic matter inputs. However, the available P concentrations along the

Chamser Kangri chronosequence were rather low relative to comparably young chronosequences (Tamburini et al.,

2012; Wang et al., 2016; Zhou et al., 2019). Moreover, the microbial P pool increasingly exceeded the available P pool

and the potential phosphatases activity increased with soil age, which can be interpreted as indications of P limitation

(Lajtha and Schlesinger, 1988). With the progressive colonization by vascular plants at sites M3 and M4 (20-50% to

70-80% plant cover, respectively, Figure S2), it is likely that microbes started to compete effectively with plants for the

available P (Lajtha and Schlesinger 1988; Seeling and Zasoski 1993; Zhou *et al.* 2013).

Contrary to what has been found previously in an alpine glacier forefield (Tamburini et al., 2012), the $\delta^{18}O_P$ of microbial

P at sites M3 and M4 were well below the isotopic equilibrium with soil water (Figure 2 and Table 2). However, the

$\delta^{18}O_P$ of microbial P at site M3 did fall within the short-term equilibrium range, which includes the isotopic value of

rainfall water. This suggests that microbes were only active under favorable temperature and moisture conditions, i.e.

after precipitations, and rapidly turned P over, bringing it to isotopic equilibrium. Under such extreme conditions,

microbial activity might be disrupted by frequent droughts and occurrence of low temperatures. The isotopic value of

microbial P might, therefore, reflect previous favorable conditions rather than those occurring at the moment sampling

(Shen et al., 2020). At site M3, this translated in a $\delta^{18}O_P$ somewhere in between the equilibrium with rain water and soil

water.

At the oldest site M4, however, the $\delta^{18}O_P$ of microbial P was much lower compared to site M3 and lied below the short-

term equilibrium range, suggesting that other processes intervened. The difference between the two sites (about 5.2‰)

might be determined by differences in soil water dynamics. The lower clay content characterizing site M3 potentially

accelerates soil water evaporation. Indeed, at the sampling time, the $\delta^{18}O_W$ at site M4 was lower by approximately 2.6‰

compared to site M3 (Table 2), thus partially explaining the offset between the two microbial P isotopic compositions.





Additionally, the hydrolysis of Po compounds induces the release of Pi with low $\delta^{18}O$ (Liang and Blake, 2006; von

     Sperber et al., 2014). Low $\delta^{18}O_P$ have been observed in microbial P of forest soils and related to phosphomonoesterase

     catalyzed reactions induced by intracellular Po recycling in response to P limiting conditions (Pistocchi et al., 2020).

     Alternatively, low metabolic activity and dormancy might induce hydrolysis of Po within cells for maintenance of basic

     functions, thus potentially leading to low $\delta^{18}O_P$ values (Lebre *et al.* 2017;Blagodatskaya and Kuzyakov 2013).

Together with differences in evaporative enrichment of soil water, differences in the metabolic status of the microbial

     community, such as internal Po recycling in response to P limitation or adverse conditions, can, therefore, explain the

     lower microbial P $\delta^{18}O$ values observed at the oldest site.

### 4.2 Contributions of biological P cycling to the inorganic P pools: long-term P dynamics

We observed a significant departure from the isotopic value of the parent material at sites M1 and M4 for available P,

     at M3 and M4 for the P bound to oxides and at M3 for mineral P (Figure 2). As the exchange of O atoms between

     phosphate and water is negligible in absence of biological processes (Blake et al., 2005; Lecuyer et al., 1999; Winter et

     al., 1940), a departure from the isotopic value of the parent material suggests that these Pi pools underwent to a certain

     extent biological transformations.

Conversely, the mineral P maintained an isotopic value similar to the parent material also at the oldest site (~100 years).

     The mineral P pool was, therefore, still composed by primary phosphate minerals after 100 years of soil development.

     This pool represented still 95% of the total P. The decline in soil mineral P stocks (-20%, see Table 1) was much less

     pronounced than that observed in other young chronosequences in alpine or other cold environment (Egli et al., 2012;

     Celi et al., 2013; Zhou et al., 2019). Unlike these studies, along the Chamser Kangri chronosequence the pH decreased

only slightly, most likely because of less acidic inputs from rainfall and a slower colonization by vascular plants, which

     prevented the rapid dissolution of primary apatite (Lajtha and Schlesinger, 1988).

     Unlike the classic Walker and Syers (1976), there was no evident decrease of total soil P in the top 0-5 cm over the first

     100 years of soil development (Table 1). However, while primary mineral P declined, an accumulation of P in organic

     and secondary mineral forms associated with metal oxides was observed, as well as a slight increase in the available P

(Table 1). The $\delta^{18}O_P$ of P bound to oxides was initially similar to that of the parent material (Figure 2, site M1), possibly

     reflecting the alteration processes of primary minerals with the formation of oxides-bound P by processes not cleaving

     the P-O bond, e.g. weathering by organic acids (Brunner *et al.* 2011; Mitchell *et al.* 2016). As this pool built up with

     soil development, its $\delta^{18}O_P$ deviated from that of the parent material and remained within or below the long-term isotopic

     equilibrium at the intermediate M3 and oldest M4 site, respectively (Figure 2 and Table 2). The exchange of phosphate

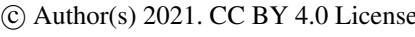

ions between the soil solution and the P bound to oxides occur within months to years (Helfenstein *et al.* 2020). The

difference in $\delta^{18}O_P$ of this pool at different sites should, therefore, depend on persistent long-term differences in the

contributing processes (e.g. sorption of phosphate from the soil solution vs phosphate from dissolution of parent material

or persistent differences in water isotopic composition).

As reported previously, $\delta^{18}O_P$ of sorbed or mineral Pi might approach isotopic equilibrium values via sorption onto soil

solid phase or precipitation as secondary minerals of biologically cycled P (Tamburini et al., 2012; Joshi et al., 2016;

Helfenstein et al., 2018; Roberts et al., 2015). In a study of a costal dunes chronosequence in New Zealand, the NaOH-

extractable Pi approached equilibrium after few millennia of soil development (Roberts et al., 2015). This has apparently

happened much earlier in the Chamser Kangri chronosequence (site M3 is approximately 50 years old). This might be

due to the relatively high proportion of the microbial P pool to the P bound to oxides, which additionally increased from

17% to 79% along the chronosequence (Table S1). Considering the large range of the long-term equilibrium (Table 2),

we cannot rule out that the P bound to oxides at this development stage (50 to 100 years) was only partially constituted

by biologically cycled P and its isotopic value only partially overprinted.

Conversely, the drop of the $\delta^{18}O_P$ value of P bound to oxides at the oldest site might be linked to the increase of Po

mineralization rate. Depleted $\delta^{18}O_P$ have been observed for NaOH-extractable Pi at the oldest sites of the mentioned

coastal dune chronosequence and related to an increasing contribution of Po mineralization to P cycling as the soil

develops (Roberts et al., 2015). The low $\delta^{18}O_P$ of P bound to oxide observed at site M4 might, therefore, result from the

sorption onto Fe and Al oxides of $^{18}O$-depleted Pi released after extracellular Po mineralization. This explanation is

corroborated by other findings, such as the sharp increase of Po pool, which tripled compared to site M3 (Table S1), the

similarly low $\delta^{18}O_P$ values of the available P and the results of the box model (see section 4.3). In the Chamser Kangri

chronosequence, this dynamic appeared to be faster than in the dune chronosequence, maybe because it affected a much

smaller quantity of the total soil P stock compared to previous studies (Roberts et al., 2015).

**4.3 Contributions of biological P cycling to the available P: short-term P dynamics**

The build-up of the microbial P pool observed across the chronosequence may suggest that this pool would increasingly

contribute to the available P. With the help of a box model, we try to elucidate the contributions to the available P from

microbial P turnover and Po mineralization. We consider that the box-model P fluxes were estimated correctly, when

the calculated $\delta^{18}O_P$ of the available P ($\delta^{18}O_{Pexpected}$, Eq. 3) closely matched the measured ones, e.g. their difference is

less than twice the standard deviation of samples replicates (0.8‰). This is the case for sites M3 and M4, when assuming

the lowest microbial P turnover time of 15 days (see Table 3). The difference between the $\delta^{18}O_{Pexpected}$ and the measured



$\delta^{18}O_P$ is instead greater for site M1 (+2.7‰) and M2 (+1.7‰). Differences between the measured and estimated $\delta^{18}O_P$

values increase for all sites with increasing microbial P turnover time. For sites M3 and M4, the estimated fluxes from

microbial P ($f_{-Pmic}$) accounted for approximately 60% to 80% of the available P. Therefore, in agreement with our initial

hypothesis, microbial P largely controlled the available P at these stages of pedogenesis (50 to 100 years). Accordingly,

the ratio of microbial P to available P increased from approximately 1 to 3 along the chronosequence. Microbial P

turnover might be accelerated by repeated freezing-thawing and drying-rewetting cycles occurring frequently in the

area, thus fostering its transfer to the available P pool, especially in a low-sorbing sandy soils (Chen et al., 2021). In

addition, microbial grazing might potentially play a role (Bernasconi et al., 2011).

At sites M1 and M2, the measured and estimated $\delta^{18}O_P$ values of the available P do not match, indicating either incorrect

assumptions or the occurrence of processes we could not account for in the box model. The $\delta^{18}O_P$ of microbial P at sites

M1 and M2 was not measured, so we assumed that these values could vary within the range of values from sites M3

and M4. However, the $\delta^{18}O_{Pexpected}$ was poorly sensitive to variations in the $\delta^{18}O_P$ of the microbial P, so we can exclude

that this assumption is a source of error. Additionally, although the $\delta^{18}O_P$ values of available P at site M2 was very close

to that of mineral P (+8.49 and +7.95‰, respectively), we exclude that fluxes from the mineral P pool could have

strongly influenced the isotopic composition of the available P. Indeed, the flux from mineral P ($f_{-Pmineral}$) should be

more than 100 times the one we estimated to account for the observed $\delta^{18}O_P$ of available P. Such a net flux appears

unlikely given the very slow exchange time (in the order of millennia) of the mineral P with the soil solution at alkaline

soil pH (Helfenstein et al., 2020). The flux from the P bound to soil oxides via sorption/desorption processes was not

accounted for in the box model. However, at the youngest sites, this flux can be considered as negligible compared to

other contributions, due to the relatively small concentration of the P bound to oxides. Moreover, the measured $\delta^{18}O_P$ of

available P at site M1 was clearly lower than that of the parent material and of the P bound to oxides (Figure 2).

As the $\delta^{18}O_{Pexpected}$ were lower compared to the measured $\delta^{18}O_P$ of available P at sites M1 and M2, we conclude that the

flux from Po mineralization ($f_{-Po}$), which would carry a low $\delta^{18}O_P$ value, was possibly overestimated. Organic P

mineralization was calculated from the $CO_2$ released by microbial respiration, assuming that a stoichiometric proportion

of Po was mineralized from non-living soil organic matter (see Eq. 4)(Achat et al., 2010; Bünemann, 2015). Stabilization

of Po compounds by adsorption onto soil particles (Zhou et al., 2019), could lead to the mineralization of a lower

proportion of Po than C. However, considering the soil sandy texture and the fact that the C:Po ratio increased along the

chronosequence, we exclude that abiotic stabilization played a major role. A second possibility is that C:Po was

overestimated. The C:Po used for the $f_{-Po}$ calculation corresponded to the C:Po of the bulk soil organic matter (Eq. 4).

However more labile Po pools, such as microbial biomass, might have a narrower C:Po ratio, for example, ranging

between 6 and 13 (Bünemann 2015 and references therein). The preferential mineralization of microbial necromass over

mineralization of the bulk soil organic matter might, therefore, explain the observed discrepancy.

At the oldest sites (M3 and M4), the good match between $\delta^{18}O_{Pexpected}$ and measured $\delta^{18}O_P$ indicates that the $f_{-Po}$ were

estimated correctly and accounted for approximately one third of total P flux to available P. Unlike the youngest sites,

Po appeared to be mineralized in stoichiometric proportion to C in the non-living soil organic matter. This finding does

not agree with the observed increase in the C:Po ratio between M3 and M4, which indicates a faster depletion of Po than

C on the pedogenesis time scale. This inconsistency might be explained by the fact that the box model only captures

short-term P dynamics, as it is built on the isotopic composition of the available P, which varies on a seasonal scale and

therefore potentially reflects transient conditions (Angert et al., 2011).

As discussed earlier, the increase of the C:Po ratio along the chronosequence suggests that in the long run Po was

mineralized faster than C. Concurrently, the colonization by vascular plants might have contributed to modify the soil

organic matter stoichiometry through inputs with higher C:P ratio. More investigation on the contribution of plant litter

to soil organic matter would be needed to clarify this aspect.

**Conclusion**

In the Chamser Kangri chronosequence, only a minor fraction of the total P in the top 5 cm of the newly formed soil

was affected by biological processes at early soil development stages. However, through a box model approach using

stocks and $\delta^{18}O$ of phosphate pools, we can conclude that the available P was mostly controlled by the microbial P via

a rapid microbial P turnover, possibly accelerated by drying-rewetting or freezing-thawing cycles. Organic P

mineralization became important in replenishing the available P pool after 50 years of soil development and contributed

also to the phosphate sorbed on oxides. Unlike previous studies in other alpine environments, the P associated to primary

minerals decreased only by 20% after approximately 100 years of soil development and its isotopic composition

reflected negligible biological cycling and secondary minerals precipitation. Finally, although cold arid conditions

slowed down the weathering of primary P minerals and controlled biological activity, microbes still played a pivotal

role in controlling the P dynamics, which affected P distribution to inorganic and organic soil pools. Our study highlights

that also in extreme environments, integrating the analysis of the isotopic composition of oxygen in soil P pools in

chronosequence studies can provide insights in the short- and long-term P dynamics.





Table 3: Box model estimation of phosphorus fluxes from the mineral, microbial and organic P (Po) pools to the

available P.

| Site | $\delta^{18}O_P$ of contributing P sources (‰) | | | P fluxes (mg m$^{-2}$ day) | | | $\delta^{18}O_P$ of available P (‰) | | |
|---|---|---|---|---|---|---|---|---|---|
| | $\delta^{18}O$ Mineral P | $\delta^{18}O_{PME}$[c] | $\delta^{18}O$ Microbial P | f-Pmineral | f-Po[d] | f-Pmic | $\delta^{18}O_{Pmeasured}$[d] | $\delta^{18}O_{Pexpected}$[e] | Difference ($\delta^{18}O_{P\ measured}$ - $\delta^{18}O_{Pexpected}$) |
| | mean (st.dev.) | - | mean (st.dev.) | - | mean (st. dev) | mean (st. dev) | mean (st. dev) | mean (st.dev) | - |
| M1 | 8.10 (0.80) | -1.74 | 10.20 (3.7) | 0.33 | 11.7 (1.8) | 2.3 (0.3) [a] | 4.66 (0.24) | 0.98 (1.10) | 3.7 |
| | | | | | | 0.4 (0.1) [b] | | -0.44 (0.78) | 5.1 |
| M2 | 7.95 (1.00) | 0.65 | 10.20 (3.7) | 0.33 | 8.1 (1.3) | 7.3 (0.8) [a] | 8.49 (0.25) | 5.60 (1.99) | 2.9 |
| | | | | | | 1.2 (0.2) [b] | | 2.71 (1.04) | 5.8 |
| M3 | 8.07 (0.71) | 0.70 | 12.85 (0.61) | 0.33 | 7.6 (1.2) | 16.7 (1.8) [a] | 8.90 (0.44) | 9.26 (0.50) | -0.4 |
| | | | | | | 2.8 (0.3) [b] | | 4.60 (0.56) | 4.3 |
| M4 | 8.72 (0.69) | 0.03 | 7.63 (0.65) | 0.33 | 5.4 (0.8) | 33.3 (3.5) [a] | 6.32 (0.65) | 6.69 (0.61) | -0.4 |
| | | | | | | 5.6 (0.6) [b] | | 4.38 (0.59) | 1.9 |

[a,b] microbial P fluxes were calculated assuming a microbial P turnover rate of 15 and 70 respectively.

[c] Oxygen isotopic composition of phosphate released by Po hydrolysis via phosphomonoesterases (Eq. 2)

[d] Estimated P flux from the mineralization of Po (Eq. 4)

[e] According to Eq. 3, mean and standard deviation were calculated with the Monte Carlo simulations

**Supporting Information**

Figure S1: Long-term temperature and moisture content in soils from January 2013 to June 2018, Figure S2: Photographs of the locations of the individual sampling sites, Figure S3: Daily soil temperature and soil moisture contents from July 2016 to June 2017 Monthly variations in temperature and moisture content in soils, Discussion S1: Water isotopes. Table S1: Sequentially extracted P pools, Table S2: Pearson correlation matrix of all measured physicochemical soil properties

**Author contribution**

Zuzana Frkova, designed the sampling strategy together with Federica Tamburini and Katerina Capkova. Zuzana Frkova. Chiara Pistocchi, Yuliya Vystavna, Katerina Capkova, Jiri Dolezal and Federica Tamburini performed the laboratory analyses. Zuzana Frkova, Chiara Pistocchi and Federica Tamburini interpreted the data. Zuzana Frkova and Chiara Pistocchi performed the statistical analyses and prepared the manuscript with contributions from all co-authors.

**Competing interests**

The authors declare that they have no conflict of interest.

**Data availability**

The dataset used in this work is available at the following link: https://doi.org/10.15454/TKOUKH

**Acknowledgment**

This study was financially supported by the Czech Science Foundation (GACR (21-26883S, GACR 21-04987S) a long-term research development project no. RVO 67985939 (Czech Academy of Sciences), Biology Centre CAS (Institute of Hydrobiology and Soil Biology) & SoWa (MEYS projects LM2015075 and EF16_013/0001782 – Soil and Water Ecosystems Research). We would like to thank Vilem Ded for assistance with graphical data analysis in R, Iva Tomkova for analyzing soils for TP and Lenka Capkova for technical help with enzyme activity measurements in the laboratory for Ecosystem Biology, at the University of South Bohemia and Institute of Hydrology in Slovakia for the analyses of stable isotopes in samples from glacier snout and glacier stream water. We are grateful to Madalina Jaggi and Stefano Bernasconi for the analysis of oxygen isotopes in soil water at the Stable Isotope Laboratory of the Geological Institute of the ETH Zurich.



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
