# Peer review of "Phosphorus dynamics during early soil development in cold desert: insights from oxygen isotopes in phosphate"

_SOIL, 2021_

## Author Response (AR1)

**REPLY TO FIRST REVIEWER**

Again, we would like to thank the reviewer very much for for positive feedback and comments. At this stage, we are requested to respond point by point to your comments. However, since most of the answers has not changed compared to the online discussion, we have highlighted in bold the ones that have changed. The lines references are referred to the new unmarked version.

- Title: I think this needs a slight re-word as it does not actually highlight which environment or the novelty/ breadth of the techniques used in this study

REPLY: Thank you for the suggestion. We modified the title as follows: "Phosphorus dynamics during early soil development in cold desert: insights from oxygen isotopes in phosphates"

- Figure 1: I think this figure could be re-worked, it would be really great to see a figure showing the sampling locations clearly, possibly with the old glacial limits as a visual representation of the speed of glacial retreat in the valley. At the moment this is very difficult to visualise from the large area map provided.

REPLY: We changed the figure according to your suggestions.

- Line 160: Five pools are mentioned here but only 4 described? Did you mean to include Po here?

REPLY: Yes, indeed the fifth pool is Po in the NaOH-EDTA extract. We modified the sentence accordingly to clarify that. Please, also note that in response to a remark from the second reviewer we changed the names of the mineral P (now HCl-P) and P bound to oxides (now NaOH-Pi) pools.

Originally: NaOH-EDTA extractable P (bound to iron- (Fe) and aluminum- (Al) oxides and to organic P)
Changed to: NaOH-EDTA extractable organic and inorganic P (NaOH-Po and NaOH-Pi, latter is supposed to contain P bound to oxi-hydroxides)

- 166: I think the working of this is a bit confusing. I read it as some sites only had values of X and Y where you mean some sites don't have values of X or Y. maybe just re phrase to make this clearer.

REPLY: We rephrased as follows:

Originally: This was the case of microbial P in the samples from sites M1 and M2 and the P bound to oxides in the sample from site M2.
Changed to: The $\delta^{18}O_P$ is missing for the microbial P pool at sites M1 and M2, and for the NaOH-Pi pool at site M2 (see Figure 2 and dataset in Frkova et al., 2021)

- Line 175: I would be surprised with a 2Hw precision of 0.1... maybe 1.0?

REPLY: We agree, there was a mistake, now corrected. The precision is 1.0 ‰ for $\delta^2H$:
Originally: "Typical precisions are better than ±0.1‰ for both $\delta^{18}O_W$ and $\delta^2H_W$, respectively".
Changed to: "Typical precisions are better than ±0.1‰ for $\delta^{18}O_W$ and ±1.0‰ $\delta^2H_W$, respectively".

- Line 179: Please add in the measurement precision here as with above.

REPLY: Sorry for this oversight. We added the following sentence:
Addition: "Reproducibility of repeated measurements of internal standards was better than ±0.06‰." **(L 197-198)**

- Lines 185-190: some info on sample weights and number of replicate analysis would be good in here as well as the average standard error of sample replicate analysis. Additionally, this error information would be good to see on Figure 2.

REPLY: thank you for pointing out this out. We added the following sentences to the paragraph:

"Samples (250-400 μg) were run in duplicates whenever possible." **(L205)**
"…and average standard deviation of samples replicates analyses was 0.5‰" **(L 209-210)**
We did not add the error on Figure 2, because we only have the analytical error from sample replicates analysis at the mass spectrometer. Indeed, because of large soil quantity needed, we could run only one

extraction per site. According to our experience with field samples, we have added a measure of the error of sample replicate extraction.

Addition: "We estimated the variability introduced by the extraction-purification procedure to be approximately ±0.5‰ according to our field samples datasets and we considered twice this value as a conservative threshold for significant differences (Pistocchi et al., 2017; Helfenstein et al., 2018; Tamburini et al., 2018)." (**L173-176**)

- How much of an issue is it for the rest of the interpretation that you only have Po from the oldest site, and how valid is it to assume that Po is the same across all of the other sites (Line 210)?

REPLY: this is a good point. Published data on the δ18OP of soil organic P and how it varies along biological and abiotic gradients are extremely rare. It is, therefore, difficult to assess the representativity of our extrapolation. We have added the following sentence to better acknowledge this issue:
"At younger sites only microbes contribute to Po, while at site M3 and M4 also residues from vascular plants, which usually are enriched in the heavy O isotope (Tamburini et al., 2018; Pfahler et al., 2013) most likely represent a source of organic P. Therefore, at younger sites, the δ18OP-org might be lower**." (L230-235)**
Additionally, we modified lines 493-496 in the discussion section:

Originally: As the δ18OPexpected were lower compared to the measured δ18OP of available P at sites M1 and M2, we conclude that the flux from Po mineralization (f-Po), which would carry a low δ18OP value, was possibly overestimated.
Changed to: As the δ18OPexpected were lower compared to the measured δ18OP of available P at sites M1 and M2, we hypothesized either that the flux from Po mineralization (f-Po), which would carry a low δ18OP value, was overestimated or that the δ18OP-org was underestimated (see Eq. 2 and 3). Since δ18OP-org was possibly overestimated at these young sites (see section 2.6), we concluded for the first option.

- Again, how representative are the full range of microbial P values going to be here. I don't question your rational in using the values from sites M3 and M4 but it would be good if you could comment on how representative you think these values are, especially as you are studying this site specifically as you expect and see major changes in soil and vegetation development leading clearly to different P pools becoming more or less important.

REPLY: as for the previous point, there are few data published on the isotopic composition of soil microbial P to allows us having an idea of the representativity of the values here. We acknowledge this issue adding the following sentence (**L268-270).**
"The used range (5.2‰) encompasses half of the natural occurring variation of δ18OP of soil microbial P from very different temperate and tropical ecosystems (+11.5 to +20.6‰, Tamburini et al., 2018)."
We additionally added a table in supplementary material (Table S4), to show how sensitive the estimations from the box model are to variations in the δ18OP of soil microbial P at site M1 and M2.

- Table 2: Maybe I'm confused here and some of the data comes from long term averages but where are the fixed values in 18Ow and temp coming from? I think the Table caption needs to make this clear.

REPLY: Indeed, captions were not so clear. The table 2 was redesigned.

- Line 311: this is a little misleading, it sounds like oxide bound P is normally similar to the parent material, but actually from your 3 data points 2 diverge and only 1 is similar. This needs re-wording.

REPLY: Sorry for that. We rephrased as follows:
Originally: The 310 P bound to Al- and Fe-oxides showed isotopic values similar to the parent material except for M3 and M4, where this pool carried a δ18OP of +15.40‰ and +6.05‰, respectively.
Changed to: Only the NaOH- Pi at site M1 showed isotopic values similar to the parent material, while at M3 and M4 this pool carried a δ18OP of +15.40‰ and +6.05‰, respectively.

- Figure 2: It would be good to see the theoretical equilibration value for the sites in a similar way to how you show PME or as a shaded bar. This will help visualise how distinct the values are from pyrophosphatase driven equilibrium. This would be good to visualise between sites, especially when

in lines 355-370 you discuss this as a potential reason for lower microbial P values between M3 and M4.

**REPLY: We changed the figure adding the equilibrium values calculated with measured soil water isotopes composition, representing the upper equilibrium limit, which allow visualising the differences between the sites. We additionally draw single analytical replicates instead of the average value to give an idea of the dispersion of the data as you suggested in a previous comment.**

- Line 370: significant suggests some statistical significance, if that's the case please quote if not just re-word. It would also be good to get a feeling of the analytical error you expect. i.e. error through chemistry as well as the MS, did you run duplicates at all through the 18Op prep?

REPLY: thank you for pointing this out. It is now explained in the manuscript that because of the amount of soil required, we could not run replicates of the extraction-purification procedure (except sample splits, n=2, for the extraction step with labelled and unlabelled HCl).
To have an idea of the analytical error expected, we used the standard deviation calculated on previous datasets from field samples (0.5‰). Although we cannot perform inferential statistics with only one field sample per site (although representative because pooled from at least ten sub-samples), we consider twice this standard deviation as a conservative estimation of a significant difference (Fay and Gerow, 2014) between samples and expected equilibrium values**. (L174-175)**

Minor comments:

REPLY: minor suggestions and comments were all implemented except the following one

- 139 cm-3 repeated?

REPLY: No, this is referred to the volumetric soil water content (cm/cm3)

**References**
Fay, D. S., & Gerow, K. (2013). A biologist's guide to statistical thinking and analysis. *WormBook: the online review of C. elegans biology*, 1-54.
Jeelani G, Deshpande R D, Shah R A & Hassan W (2017) Influence of southwest monsoons in the Kashmir Valley, western Himalayas, Isotopes in Environmental and Health Studies, 53:4, 400-412, DOI: 10.1080/10256016.2016.1273224

**REPLY TO SECOND REVIEWER**

We would like to thank you again for your detailed and constructive comments and for supporting the publication of our paper. Please find below our detailed reply to each of your comment. Most of the replies remain unchanged compared to the online discussion, we have highlighted in bold the changes made to our replies. Please, note that the reference to the lines apply to the unmarked manuscript version.

Specific comments:

Three major issues should be addressed in the revision of the paper:

Assignment of Hedley fractions to soil minerals. Recent research has shown that the assignment of different Hedley P fractions to specific mineral types is not straightforward, and in specific cases may be completely wrong. See: Gu & Margenot (2021) Plant Soil 459:13–17 (https://doi.org/10.1007/s11104-020-04552-x) and Klotzbücher et al. (2019), J Plant Nutr Soil Sci 182:570-577. https://doi.org/10.1002/jpln.201800652.

Considering this, the authors should be more careful in assigning their Hedley P fractions to specific mineral phases (e.g. "Al- and Fe-bound" phases). This is particularly the case because (i) no support of their statements by other analyses (e.g. P K-edge XANES), (ii) not even any information about the absence, presence, and (iii) no data on contents of different potentially P-sorbing Al- and Fe minerals have been provided in the paper.

REPLY: Thank you for raising this point. Indeed, in this study, we had no independent evidence of the specific composition of the Hedley P pools. Accepting your suggestion, in the new manuscript version we have changed the P pools names and called them by the name of the extraction reagent (e.g. HCl-P). We did not change the names of the microbial and available P as the first represents a biological fraction and the resin P is usually considered a very good proxy for the available P. We changed the description of these pools in the materials and methods accordingly (lines 177-179):

Originally: "NaOH-EDTA extractable P (bound to iron- (Fe) and aluminum- (Al) oxides and to organic P), and HCl extractable P (mineral P, mostly P bound to Ca in apatite)"

Changed to: "NaOH-EDTA extractable organic and inorganic P (NaOH-Po and NaOH-Pi, latter is supposed to contain P bound to oxi-hydroxides), and HCl extractable P (HCl-P, targeting mostly P bound to Ca)."

I understand that conduction of XANES analyses is probably out of reach for the authors of the paper, but analytical determination of dithionite-citrate-bicarbonate Fe (Fed, estimating Fe present in well-crystallized Fe oxyhydroxides, like goethite) and of acidic oxalate-extractable Fe and Al (Feo, Alo, estimating the Al and Fe present in short-range order minerals and gibbsite) may help to support the assignment of the NaOH-extractable Hedley P fraction to Al and Fe minerals. To be on the safe side, of course, one has to refrain from attributing the Hedley fractions to particular minerals as a whole, and just focus on the different availability of the different fractions to plants and soil microorganisms. If I understand the key message of the paper correctly, this is the main aim of the paper, and attribution of the Hedley fractions to particular mineral phases is of secondary importance.

REPLY: As you mentioned, the composition of Hedley fractions in terms of mineral phases, although interesting, is of secondary importance in our paper, as we focussed on biological availability of these pools as revealed by their isotopic composition. As replied above, we have changed the names of the Hedley pools throughout the manuscript in a way that is more "neutral".

We have down-tuned in the interpretation of P pools data in the discussion, see for examples lines 491-493:

Originally: "However, at the youngest sites, this flux can be considered as negligible compared to other contributions, due to the relatively small concentration of the P bound to oxides."

Changed to: "However, at the youngest sites, this flux can be considered as negligible compared to other contributions, due to the relatively small concentration of the NaOH-Pi, which is supposed to target P bound to oxides."

And lines 527-529:

Originally: "50 years of soil development and contributed also to the phosphate sorbed on oxides"

Changed to: "50 years of soil development and contributed also to the phosphate sorbed onto secondary minerals, presumably Fe and Al oxides"

2) I strongly recommend analysis of some additional soil variables, provided that some sample material is still available. (1) Analysis of dithionite-citrate-bicarbonate Fe (Al) and acidic oxalate-extractable Fe and Al (as mentioned before) would help to clarify the assignment of the reported Hedley P fractions to mineral phases. Moreover, it is a generally important soil variable, and helps to characterize the different soils in the study of Frkova et al. with respect to their stage of pedogenesis. I assume that some Alo and Feo will be present particularly in the older soils of the chronosequence, even though the pH is >7.7 (which normally prevents silicate weathering). This may raise discussions about the sources of pedogenic oxides (see an earlier paper of mine on two glacier forefields in China (also Tibetian Plateau) and Switzerland (Damma): Prietzel et al. 2013, GCA 108:154-171; https://doi.org/10.1016/j.gca.2013.01.029). Alternatively, Alo and Feo also includes organically bound Fe and Al in addition to/instead of mineral-bound Fe and Al – However, this line of argument may disprove the statement made in the paper that the NaOH-extractable P is bound to Al and Fe minerals.

REPLY: Thank you for this suggestion. Unfortunately, no soil material is left from the campaign of 2017. However, we performed a Fe sequential extraction on older samples (2011) taken in the proximity of our four sites and kept at -20°C. However, in the shipping the sample from site 4 was lost, so no analyses could be done.

We added these data in table S3 (supplementary materials) and commented them in the discussions section:

Addition: The low amount of Fe associated to oxides (Table S3) compared to other young chronosequence also indicates a slow soil development (Prietzel et al., 2013).

Addition: "Accordingly, Fe associated to poorly ordered oxides increased between the youngest sites M1 and the intermediate sites M2-3 (Table S3)."

We did not discuss further the sources of Fe oxides as the site M4 was missing and we considered such discussion beyond the scope of the paper.

Moreover, I recommend measuring inorganic carbon (carbonate) and the electric conductivity in the different soil samples. The climate conditions at the study sites, as well as the high pH in the investigated soils (7.7 – 8.7) both indicate the presence of carbonate and /or salt accumulation in the topsoil. Additionally, the good correlation between pH and total K in the different topsoils (see Table 1) suggests salt accumulation, which has a strong influence on weathering, soil P speciation, and probably also soil microbial communities and activity. The EC values are a good indicator for salt accumulation, and thus should be analyzed. I suspect

that EC values are increased in the studied soils compared to ordinary soils under humid climate, and the investigated soils thus are probably affected by topsoil salt accumulation, which may be temporarily or continuously present at varying levels. If the investigated soils turn out to be affected by salt accumulation due to the arid-cool climate, the influence on weathering, soil P speciation, soil microbial communities, and activity should be addressed more deeply in the paper.

REPLY: As we previously replied we have not performed EC ad carbonates analyses on our soil samples. However, in (Aschenbach et al., 2013) carbonates in lateral moraines of the Chamser Kangri glacier were reported to vary between undetectable and 4.3%, therefore rather low. Unfortunately, we could not retrieve the raw salinity data from previous sampling campaigns. Dolezal et al., (2016) report that plant communities in this environment are composed with species with affinities to soil salinity, so we can reasonably assume that microbial communities as well are adapted to some extent to soil salinity. As the measured O isotopes in phosphate in microbial and available P seem to indicate, such communities are active in cycling P when conditions are favourable. We therefore think that the salinity data, although useful would not change the interpretation of our data. Unfortunately, we cannot go much further in our discussion in terms of the effect of salt on microbial communities, but we consider this beyond the scope of the paper.

3) In this respect I recommend reading a recent paper of mine dealing with P speciation changes in cold arid glacier forefield regions of Antarctica (Prietzel et al., 2019, GCA 246:339-362. https://doi.org/10.1016/j.gca.2018.12.001 and the references therein. I have the impression that the environmental conditions in the paper of Frkova et al. and those reported in my 2019 study are quite similar in many (aridity, high UV influence) but not all (seasonality, day length) aspects.

REPLY: Thank you for suggesting this reference. As you pointed out earlier, there are only few published data on P dynamics in cold-arid environment. We have integrated the reference in the introduction and in the discussion (L 85).

Originally: "Direct forefield observations on the role of microorganisms in P cycling using O isotopes in phosphate are rare."

Changed to: "Direct forefield observations on the role of microorganisms in P cycling are rare especially under cold arid conditions (Prietzel et al., 2019)."

And L 421-424:

Addition: "However, these environments are sensibly more humid than the Chamser Kangri forefield. As reported by Prietzel et al., (2019) for polar cold arid soils, the weathering of primary P minerals like apatite is strongly retarded by the lack of water, which slows down soil acidification."

One minor issue that in a soil science paper I would like to see some soil type (WRB) description, maybe also horizon designations for the studied topsoil horizons.

REPLY: According to Gupta and Arora (2017) the soils of the Ladakh region are mostly classified as Entisols (USDA classification). According to our field knowledge, they would fall in the order of Leptosols (WRB classification).

Addition (line 125): "They can be classified as Leptosols according to the WRB (Anon, 2006)."

And lines 137-138:

Originally: "Soil samples were collected along the frontal (M1-3) and lateral (M4) moraines of the Chamser Kangri glacier chronosequence (5711, 5710, 5700 and 5598 m a.s.l., M1-M4 respectively)."

Changed to: "Soil samples were collected along the frontal (M1-3) and lateral (M4) moraines of the Chamser Kangri glacier chronosequence (5711, 5710, 5700 and 5598 m a.s.l., M1-M4 respectively) from the A1 horizon."

Technical corrections

L29: Please specify soil depth or horizon, where 95% of total P is mineral P

REPLY: It is now specified at L29

L30: Can you specify the "primary phosphate minerals"?

REPLY: Done (L30)

L34: should read: "becomes" instead of "become"

REPLY: Done

L149-152: Please specify: Have the analyses been conducted on sieved or in ground samples?

REPLY: All the analyses have been conducted on sieved samples (<2mm), except for the 18Op in the parent material. For that we milled a sample of the parent material and then dissolved it in HCl (**L154-155 and 179**).

L151: K , Mg, and Ca are not micronutrients

REPLY: You are right. We corrected the mistake

L152: This is "pseudo-total" P rather than total P, because silicates are not completely dissolved by HNO3/HClO4 digestion and the P bound in silicates thus is probably underestimated.

REPLY: Thank you for pointing this out. We modified the sentence as follows (**L 161-162**):

Originally: "total P and other major elements were analyzed by ICP-OES after sequential digestion by HNO3 and HClO4 (Kopacek et al., 2001)."

Changed to: "total P and other major elements were analyzed by ICP-OES after digestion by HNO3 and HClO4, although total P might be underestimated because of incomplete dissolution of silicates during the digestion (Kopacek et al., 2001)."

L160: How can NaOH-extractable P be bound to organic P? Please reword sentence in bracket

REPLY: The sentence was rephrased (L177-178):

Originally: "NaOH-EDTA extractable P (bound to iron- (Fe) and aluminum- (Al) oxides and to organic P)"

Changed to: "NaOH-EDTA extractable organic and inorganic P (NaOH-Po and NaOH-Pi, latter is supposed to contain P bound to oxi-hydroxides)"

L257: The estimation of bulk density should be described in more detail

REPLY: The equation has been added and the reference changed, the current reference is accessible and reports the equation we used originally from Leonaviciute, 2000, which was not accessible (L166-167):

Originally: "was estimated using a pedotransfer function (Leonaviciute, 2000), corresponding to eluvial deposits considering both the soil texture and organic carbon content"

Changed to: "was estimated using a pedotransfer function (BD = 1.70398 - 0.00313 Silt + 0.00261 Clay 0.11245 Organic carbon, Abdelbaki, 2018), corresponding to eluvial deposits"

L259 "0.03 to 0.6" Please add unit also here

REPLY: Sorry for the oversight. The units have been added

L287: Can you estimate average evaporation and a water balance from the d-excee data? Would be nice

REPLY: we can try an approximation, but it will be rather qualitative. The climate is a cold desert and evaporation signal is very biased. We will fully respond later on to this comment.

L301: Please report mineral P content in addition to percentage

REPLY: done

L303/4: Please report percentages of total P in addition to P content data

REPLY: The information has been added

L321: should read: "Nutrient" instead of "Nutrients"

REPLY: Done

L325: should read: "total topsoil N concentration"

REPLY: Done

L348: Should read: "precipitation events" instead of "precipitations"

REPLY: Done

L377: Important: These environments are much more humid. See my specific comment #2

REPLY: We modified the whole paragraph (**L421-425**).

Originally: Unlike these studies, along the Chamser Kangri chronosequence the pH decreased only slightly, most likely because of less acidic inputs from rainfall and a slower colonization by vascular plants, which prevented the rapid dissolution of primary apatite (Lajtha and Schlesinger, 1988).

Changed to: However, these environments are sensibly more humid than the Chamser Kangri forefield. As reported by Prietzel et al., (2019) for polar cold arid soils, the weathering of primary P minerals like apatite is strongly retarded by the lack of water, which slows down soil acidification. Along our chronosequence, this effect together with the slow colonization by vascular plants prevented the rapid dissolution of primary apatite (Lajtha and Schlesinger, 1988).

L379: Maybe change to "which slowed down soil acidification, and prevented…"

REPLY: See above

L402: Should read: " Depleted d18Op values have been observed"

REPLY: Modified

L406: Replace " findings" by "soil features" or "soil properties"

REPLY: Done

L413: Contributions (of what?). Please specify

REPLY: We have changed the title of the section as follows: "Contributions of microbial P turnover and Po mineralization to the available P: short-term P dynamics"

L424: Should read: "in low-sorbing sandy soils" instead of "in a low-sorbing sandy soils"

REPLY: Modified

L430: Should read: "d18Op value" instead of "d18Op values"

REPLY: Modified

L467: Maybe add: (alpine environments) with humid climate

REPLY: Modified as suggested

**References**

Aschenbach, K., Conrad, R., Řeháková, K., Doležal, J., Janatková, K., and Angel, R.: Methanogens at the top of the world: occurrence and potential activity of methanogens in newly deglaciated soils in high-altitude cold deserts in the Western Himalayas, Front. Microbiol., 4, 359, https://doi.org/10.3389/fmicb.2013.00359, 2013.
Dolezal, J., Dvorsky, M., Kopecky, M., Liancourt, P., Hiiesalu, I., Macek, M., Altman, J., Chlumska, Z., Rehakova, K., Capkova, K., Borovec, J., Mudrak, O., Wild, J., and Schweingruber, F.: Vegetation dynamics at the upper elevational limit of vascular plants in Himalaya, Sci. Rep., 6, 24881, https://doi.org/10.1038/srep24881, 2016.